# The Impact of Climate Change on Agricultural Insect Pests

**DOI:** 10.3390/insects12050440

**Published:** 2021-05-12

**Authors:** Sandra Skendžić, Monika Zovko, Ivana Pajač Živković, Vinko Lešić, Darija Lemić

**Affiliations:** 1Department of Agricultural Zoology, Faculty of Agriculture, University of Zagreb, Svetosimunska 25, 10000 Zagreb, Croatia; ipajac@agr.hr (I.P.Ž.); dlemic@agr.hr (D.L.); 2Department of Soil Amelioration, Faculty of Agriculture, University of Zagreb, Svetosimunska 25, 10000 Zagreb, Croatia; mzovko@agr.hr; 3Innovation Centre Nikola Tesla, Unska 3, 10000 Zagreb, Croatia; vinko.lesic@icent.hr

**Keywords:** climate change, global warming, food security, agriculture, insect pests

## Abstract

**Simple Summary:**

Climate change and extreme weather events have a major impact on crop production and agricultural pests. As generally adaptable organisms, insect pests respond differently to different causes of climate change. In this review, we address the effects of rising temperatures and atmospheric CO_2_ levels, as well as changing precipitation patterns, on agricultural insect pests. Since temperature is the most important environmental factor affecting insect population dynamics, it is expected that global climate warming could trigger an expansion of their geographic range, increased overwintering survival, increased number of generations, increased risk of invasive insect species and insect-transmitted plant diseases, as well as changes in their interaction with host plants and natural enemies. As climate change exacerbates the pest problem, there is a great need for future pest management strategies. These include monitoring climate and pest populations, modified integrated pest management strategies, and the use of modelling prediction tools which are presented here.

**Abstract:**

Climate change and global warming are of great concern to agriculture worldwide and are among the most discussed issues in today’s society. Climate parameters such as increased temperatures, rising atmospheric CO_2_ levels, and changing precipitation patterns have significant impacts on agricultural production and on agricultural insect pests. Changes in climate can affect insect pests in several ways. They can result in an expansion of their geographic distribution, increased survival during overwintering, increased number of generations, altered synchrony between plants and pests, altered interspecific interaction, increased risk of invasion by migratory pests, increased incidence of insect-transmitted plant diseases, and reduced effectiveness of biological control, especially natural enemies. As a result, there is a serious risk of crop economic losses, as well as a challenge to human food security. As a major driver of pest population dynamics, climate change will require adaptive management strategies to deal with the changing status of pests. Several priorities can be identified for future research on the effects of climatic changes on agricultural insect pests. These include modified integrated pest management tactics, monitoring climate and pest populations, and the use of modelling prediction tools.

## 1. Introduction

Throughout history, human population growth has been accompanied by many changes in everyday life, culture, technology, science, the economy, and agricultural production. Agricultural production has also undergone many major changes—agricultural revolutions—which have influenced by the development of civilization, technology, and general human advancement. However, the exceptional population growth in the last 100 years has had many undesirable consequences that (along with changes in environmental conditions) impact the security of the food supply. The growing world population has rising demands for crop production and accordingly, by 2050, global agricultural production will very likely need to be doubled to meet that kind of increasing demand [1]. For food security, numerous studies have recommended that enhancing crop yield, rather than clearing more land surface for crop production, is the most sustainable approach [2]. Modern scientific research and agronomy are focused on climate change and related phenomena—rising global temperature and atmospheric carbon dioxide concentrations, heat waves, flooding, intense storms, droughts and other extreme weather events. Therefore, more attention to the aforementioned abiotic factors is given in agricultural science, as the tendency to reduce yield loss due to such conditions increases. Regarding crop production, changes in precipitation patterns may potentially have higher importance than temperature rise, especially in areas where dry seasons present a limiting factor for agricultural production [3]. One of the major biotic factors are pests, which are also impacted by climate change and weather disruptions. Temperature rise directly affects pest’s reproduction, survival, spread and population dynamics as well as the relationships between pests, the environment, and natural enemies [4]. As such, it is very important to monitor pest’s appearance and abundance as the conditions of their occurrence can change at a high pace. This paper will review the impact of some of the predicted climate changes, especially the rise of atmospheric carbon dioxide concentrations and temperatures along with changeable precipitation pattern effects on the biology and ecology of harmful insects, especially invasive pest species, which can be a major problem in crop production. Potential solutions for the current issues in plant production will be presented, mostly in the form of modified integrated pest management (IPM) strategies, which include IPM and the production of healthy food in an environmentally friendly way as well the monitoring techniques and modelling prediction tools.

## 2. Climate under Change

The climate is a crucial element that determines various characteristics and distributions of managed and natural systems, including hydrology and water resources, cryology, marine and freshwater ecosystems, terrestrial ecosystems, forestry and agriculture [5]. It can be explained as the phenomenon that involves changes in environmental factors such as temperature, humidity and precipitation over many years. As a result of increased temperatures, climate extremes, increased CO_2_ and other greenhouse gases (GHGs) as well as altered precipitation patterns, global food production is under severe threat [6]. Global warming is a serious problem facing the world today. It has reached record breaking levels as evidenced by unprecedented rates of increase in atmospheric temperature and sea level [7]. According to the World Meteorological Organisation (WMO), the world is now about one degree warmer than before widespread industrialization. The Intergovernmental Panel on Climate Change (IPCC) [7] also reported that each of the last three decades has been increasingly warmer, with the decade of the 2000s being the warmest. Based on a range of global climate models and development scenarios, it is expected that the Earth could experience global warming of 1.4 to 5.8 °C over the next century [8]. The main cause of global warming is increased concentrations of greenhouse gases in the atmosphere. The most prevalent atmospheric gases are carbon dioxide (CO_2_), methane (CH_4_), and nitrous oxide (N_2_O), which are caused by many anthropogenic activities including burning off the fossil fuels and land-use change [9]. Looking at the period of industrialization in the last two centuries, the concentration of greenhouse gases has increased immensely compared to the pre-industrial era [10]. Among the greenhouse gases, CO_2_ is the most important and the most abundant [11]. The increase in atmospheric CO_2_ is one of the most recorded global changes in the atmosphere in the last half century [12]. Its concentration in the atmosphere has increased dramatically to 416 ppm, against 280 ppm reported from pre-industrial period, and is likely to double in 2100 [8,13]. CO_2_ is considered a greenhouse gas due to its high absorbance in certain wavelengths of thermal infrared radiation emitted from the Earth’s surface. The greater the amount of atmospheric gases that absorb thermal infrared radiation from the Earth’s surface, the greater the proportion of radiation emitted from the atmosphere toward the Earth’s surface [14]. As a result, the long-wave balance of the Earth’s surface becomes less negative, while more energy is available for sensible and latent heat flux at the Earth’s surface. As more energy is available for heat flux, this leads to an increase in air temperature [15]. Changes in extreme weather and climate events have been observed since the mid-20th century. Many of these changes, which have been linked to anthropogenic influences, include a reduction in cold temperature extremes, an increased occurrence of warm temperature extremes, enhanced rates of sea level rise, and an increase in the frequency of heavy precipitation events in numerous regions. Heat waves are expected to become more frequent and last longer, and extreme precipitation events are expected to be more intense and frequent in certain areas [7]. It is very likely that the precipitation pattern will change and not be uniform. In the higher latitudes and equatorial Pacific, there appears to be an increase in mean annual precipitation. In the dry mid-latitude and subtropical regions, mean precipitation is likely to decrease, while in the wet mid-latitude regions, mean precipitation is likely to increase. Extreme precipitation events in most mid-latitude areas and humid tropical regions are likely to become more frequent and intense [7]. The United Nations (UN) and the IPCC have made numerous decisions to reduce GHGs emissions, provide financial assistance to developing countries, and improve adaptive capacity to meet the challenges posed by the harmful effects of climate change.

### 2.1. Impact of Climate Change on Crop Production

Agriculture, often referred to as an open-air factory, is an economic activity that depends heavily on climate and certain weather conditions to produce food and many other goods necessary to sustain human needs. Moreover, agriculture is an activity that is exceptionally vulnerable to climate change and the impacts of climate change are characterized by various types of uncertainty [16]. Climate change is estimated to have both positive and negative impacts on agricultural systems at the global level, with negative impacts outweighing the positive ones [17]. Temperature increases, altered precipitation patterns, and increased CO_2_ concentrations have a significant impact on ecosystems, ranging from species to ecosystem levels [18]. For this review, it is important to first explain the effects of climate change on crop production, as the effects of climate change on insect pests depend on the plant species on which these insects thrive and feed.

#### 2.1.1. Impact of Temperature Increase

Temperature is considered one of the most important factors affecting the distribution and abundance patterns of plants due to the physiological limits of each species [19]. It is a factor that limits the geographical areas where different crops can be grown, as well as a factor that affects the rate of development, growth and crop yields [20]. Agricultural crops have basic temperature requirements to complete a particular phenophase as well as the entire life cycle. In addition, extremely low and high temperatures can have detrimental effects on crop development, growth, and yield, especially during critical phenophases (such as anthesis) [21]. It is predicted that the spring–summer season will have higher air temperatures, which would be beneficial for crop production in northern locations where the length of the growing season is currently a limiting factor [22]. The effects of temperature increase are generally associated with other environmental factors such as water availability, the occurrence of strong winds, and the intensity and duration of sunlight [20]. The direct negative temperature influence on yield could be additionally impacted by indirect temperature influences on these environmental factors. For example, the rise in temperature increases atmospheric water demand, which may lead to additional water stress due to higher water pressure deficits, which subsequently reduces soil moisture and eventually decreases yield [23]. Other indirect effects of temperature rise include increased frequency of heat waves and the impacts on pests, weeds, and plant diseases [7].

#### 2.1.2. Impact of Elevated CO_2_ Concentration

CO_2_ is the essential chemical compound for photosynthesis, a process in which water and CO_2_ are converted into sugars and starch, powered by solar energy. Photosynthesis occurs in the green pigments of leaves, and CO_2_ must enter through stomatal openings [24]. Since carbon is the key element in the structure of the plant, increased CO_2_ concentration enables faster growth due to rapid carbon assimilation [25]. The main effects of elevated CO_2_ on plants include a reduction in transpiration and stomatal conductance, improved water and light-use efficiency, and thus an increase in photosynthetic rate. Consequently, elevated atmospheric CO_2_ concentrations could have a direct impact on ecosystems by stimulating plant development and growth [26]. Although higher CO_2_ concentrations could increase crop yields, the magnitude of the effect remains to be determined. Finally, CO_2_ will affect C3 and C4 plants differently because the C4 group of plants are less sensitive to an increase in atmospheric CO_2_ concentration than C3 plants [22,27]. However, Rötter and Van de Geijn [24] found that both C3 and C4 plants will benefit from an increase in atmospheric CO_2_ concentration. The vast majority of crop plants use the C3 carboxylation pathway (Calvin cycle), which is the oldest of the carbon fixation pathways and is found in plants of all taxonomies [28,29]. The concept of C3 photosynthesis is based on the observation that the first product of photosynthesis is a 3-carbon molecule, whereas in C4 photosynthesis the first photosynthetic product is a 4-carbon molecule. C4 photosynthesis occurs in the more highly developed plant taxa and the major C4 plant species include maize, sorghum and sugarcane, all of tropical origin [24]. Only 3% of all flowering plant species are C4 plants and yet they account for about 50% of the 10,000 grass species [30]. C4 plants have about 50% higher photosynthetic efficiency than C3 plants (e.g., rice, wheat, soybean, potato, etc.), indicating that their productivity is very high. This is due to the different mechanism of carbon fixation by the two photosynthesis types. The C3 type of photosynthesis uses only the Calvin cycle for CO_2_ fixation, which is catalyzed by the Rubisco enzyme that takes place inside the chloroplast in the mesophyll cell. In the C4 group of plants, photosynthetic activities are divided between mesophyll and bundle sheath cells (BS), which are biochemically and anatomically distinct. Primary carbon fixation is catalyzed by the enzyme PEPC (phosphoenolpyruvate carboxylase), which forms OAA (oxaloacetate) from CO_2_ and PEP (phosphoenolpyruvate). OAA is converted to the salt of malic acid (malate) and then disperses into the BS cell where it is decarboxylated to provide an elevated CO_2_ concentration around the Rubisco enzyme. Finally, the initial substrate of the C4 cycle, PEP, is regenerated in the mesophyll cell by the enzyme PPDK (pyruvate orthophosphate dikinase) [31]. The mechanism of CO_2_ concentration suppresses the oxygenation reaction by Rubisco and the subsequent energy-wasting process of photorespiration, resulting in increased photosynthetic yield and improved efficiency of water and nitrogen use compared to C3 plants [32]. C4 plants are usually found in warmer environments, such as tropical grasslands, where photorespiration rates would be very high for C3 plants [30]. Therefore, under these conditions, the efficiency of C4 photosynthesis is greater than that of C3 photosynthesis [29]. Cure et al. [33] also noted that plants with nitrogen-fixing symbionts (e.g., soybean, alfalfa, lupine, etc.) tend to benefit more from an increased CO_2_ resource than other plant species under favourable environmental conditions for both the plant and the symbiont.

#### 2.1.3. Impact of Changeable Precipitation Pattern

Crop production is strongly influenced by water availability. Climate change will alter rainfall patterns, soil moisture storage, evaporation and runoff. It is estimated that more than 80% of total global crop production is supplied by rainfall, and therefore changes in total seasonal rainfall or its patterns are very important [34]. There is clear evidence of amplification of the global hydrological cycle, which is strongly influenced by changes in temperature. However, its impact on crop production is still very difficult to predict as it depends on other climate parameters such as the intensity and frequency of extreme weather events [35]. Changes in precipitation patterns may be of greater importance to agriculture than changes in temperature, especially in regions where dry seasons may be a limiting factor for crop production [3]. According to Lickley and Solomon [36] a drying trend is emerging in Southern and Northern Africa, parts of Latin America, Australia and Southern Europe. Moreover, the models predict significant drying for these regions as well as for the southern parts of North America by mid-century, with an increase in drought of more than 10% and a moisture deficit of more than 200 mm per year. In Mediterranean countries, cereal yields are limited by water scarcity, heat stress and short grain filling duration. Therefore, permanent crops such as olives, grapevine and citrus are of greater importance in this region. These crops are greatly affected by extreme weather events such as hail and storms, which can subsequently reduce or completely destroy yield [34]. Due to high evapotranspiration and limited rainfall, attention should be given to the development of irrigation techniques that allow efficient and effective use of available water resources, as well as good agronomic practices that emphasize moisture conservation and thus improve crop productivity [37]. Lack of water in the soil can cause plants to lose their biological functions and become even more susceptible to diseases and pests [38]. On the other hand, the world has become wetter in large areas such as northern Europe and eastern parts of the Americas, with extreme rainfall events contributing strongly to the increase in global precipitation [8]. Direct analysis of precipitation extremes (largest annual 1-day precipitation accumulation/largest annual 5-day precipitation accumulation) shows that extreme precipitation has increased in large parts of the world, with an increase in the potential of a typical 2-year event of about 7% over the period from 1951 to 1999 [39,40]. Due to the wet weather conditions on the Atlantic coast and in the European mountainous regions, there are cold and rainy summers that lead to yield and quality losses in various arable crops [34]. These wet conditions can also affect the workability of the soil and reduce the number of working days of agricultural machinery [41]. Overall, the exact nature of forthcoming climatic changes is still uncertain, but current projections indicate that they are very likely to have serious impacts on crops in the near future.

### 2.2. Impact of Climate Change on Insect Pests

Global climate changes have significant impacts on agriculture and also on agricultural insect pests. Agricultural crops and their corresponding pests are directly and indirectly affected by climate change. Direct impacts are on pests’ reproduction, development, survival and dispersal, whereas indirectly the climate change affects the relationships between pests, their environment and other insect species such as natural enemies, competitors, vectors and mutualists [4]. Insects are poikilothermic organisms; the temperature of their body depends on the temperature of the environment. Thus, temperature is probably the most important environmental factor affecting insect behaviour, distribution, development and reproduction [42]. Therefore, it is very likely that the main drivers of climate change (increased atmospheric CO_2_, increased temperature and decreased soil moisture) could significantly affect the population dynamics of insect pests and thus the percentage of crop losses [43]. Climate change creates new ecological niches that provide opportunities for insect pests to establish and spread in new geographic regions and shift from one region to another [44]. The complexity of physiological effects exerted by rising temperatures and CO_2_ can profoundly affect interactions between agricultural crops and insect pests [45,46,47]. Therefore, farmers can expect to face new and intense pest problems in the coming years due to the changing climate. The spread of crop pests across physical and political boundaries threatens food security and is a global problem common to all countries and all regions [44].

#### 2.2.1. Response of Insect Pests to Increased Temperature

Insect physiology is very sensitive to changes in temperature, and their metabolic rate tends to approximately double with an increase of 10 °C [48]. In this context, many researchers have shown that increased temperature tends to accelerate insect consumption, development, and movement, which can affect population dynamics by influencing fecundity, survival, generation time, population size, and geographic range [49]. Species that cannot adapt and evolve to increased temperature conditions generally have a difficult time maintaining their populations, while other species can thrive and reproduce rapidly. Temperature plays an important role in metabolism, metamorphosis, mobility, and host availability, which determines the possibility of changes in pest population and dynamics [6] (Figure 1). From the distribution and behavior of contemporary insects, it can be hypothesized that rising temperatures should be accompanied by increased herbivory [50]. Given the distribution and behaviour of insect pests, it can be hypothesised that an increase in temperature should be associated with increased herbivory [50], as well as changes in the growth rate of insect populations [51]. Thus, insect populations in tropical zones are predicted to experience a decrease in growth rate as a result of climate warming due to the current temperature level, which is already close to the optimum for pest development and growth, while insects in temperate zones are expected to experience an increase in growth rate [51]. The same authors confirmed this theory by estimating changes in the growth of pest populations in the production of the world’s three major grain crops (wheat, rice and maize) under different climate change scenarios. According to the study, for wheat, which is normally grown in temperate climates, warming will accelerate the growth of pest populations. For rice grown in tropical zones, they predict a decrease in the growth of pest populations, and for maize grown in both temperate and tropical regions, mixed responses to the growth of pest populations could be expected [51].

The effects of increased temperatures are greater for aboveground insects than for those that spend most of their life cycle in the soil, because soil is a thermally insulating medium that can buffer temperature changes and thus reduce their impact [49]. For example, under warmer conditions, aphids are less susceptible to the aphid alarm pheromone they normally release when threatened by insect predators and parasitoids, which can lead to increased predation [52]. Whitefly populations are primarily regulated by environmental factors such as temperature, precipitation, and humidity in general. High temperature along with high humidity correlates positively with whitefly population build-up [53].

Future changes in insect population dynamics depend on the level of global temperature increase in coming years. Climate models predict that the average temperature of the globe will increase by 1.8–4 °C by the end of the current century [54,55,56]. As ambient temperatures generally increase toward optimal temperatures for growth and development of many insect pest species, potentially reducing thermal constraints on population dynamics, the severity of pest infestations is expected to increase under global warming scenarios [57]. However, given the narrow ecological niche requirements, physiological tolerances of insects, and variable effects of temperature on their phenology and life history, global warming may not uniformly increase pest abundance and thus economic crop losses [58]. In their analysis, Lehmann et al. [58] showed mixed responses to climate warming in different insect pest species. The results of their analysis indicate that temperature rise leads to increased pest severity in most of their insect case studies. However, 59% of all species analysed showed responses that could reduce their harmful impact, mostly via reduced physiological performance and range contraction.

Another study of about 1100 insect species found that climate change due to global warming will drive about 15–37% of these species to extinction by 2050 [59,60].The general consequences of global warming on insect dynamics include: expansion of geographic range, increased survival rates of overwintering populations, increased risk of introduction of invasive insect species, increased incidence of insect-transmitted plant diseases due to range expansion and rapid reproduction of insect vectors, reduced effectiveness of biological control agents such as natural enemies, etc.

#### 2.2.2. Response of Insect Pests to Increased CO_2_ Concentration

Elevated concentrations of atmospheric CO_2_ can affect the distribution, abundance, and performance of herbivorous insects. Such increases can affect consumption rates, growth rates, fecundity, and population densities of insect pests [61]. Currently available data suggest that the effect of elevated atmospheric CO_2_ on herbivory is not only highly specific to individual insect species, but also to particular insect pest–host plant systems [62]. The effects of increasing CO_2_ levels on insect pests are highly dependent on their host plants. Increased CO_2_ levels would have a greater impact on C3 crops (wheat, rice, cotton, etc.) than on C4 crops (corn, sorghum, etc.). Therefore, these differential effects of elevated atmospheric CO_2_ on C3 and C4 plants may result in asymmetric effects on herbivory, and the response of insects feeding on C4 plants may differ from that of C3 plants. C3 plants are likely to be positively affected by elevated CO_2_ and negatively affected by insect response, whereas C4 plants are less responsive to elevated CO_2_ and therefore less likely to be affected by changes in insect feeding behavior [63].

As mentioned in the previous section, increased CO_2_ levels are likely to affect plant physiology by increasing photosynthetic activity, resulting in better growth and higher plant productivity. This in turn would indirectly affect insects by changing both the quantity and quality of plants and vegetation. A common feature of plants grown under elevated CO_2_ is a change in the chemical composition of leaves, which could affect the nutrient quality of foliage and palatability to leaf-feeding insects [64] (Figure 2). In addition, such crops often accumulate sugars and starches in their leaves, which reduces palatability by altering the C (carbon) to N (nitrogen) ratio [65]. Nitrogen is the key element in the insect’s body for its development, and therefore increased CO_2_ concentration leads to increased plant consumption rate in some pest groups [66]. This can lead to increased levels of plant damage, as pests must consume more plant tissue to obtain an equivalent level of food. Increased consumption rates are a common response in foliage feeders, such as caterpillars, miners, and chewers, to a reduction in nitrogen, as predicted by CO_2_ fertilization, with compensatory feeding [50,67]. Hamilton et al. [67] conducted an experiment in which soybeans were grown at elevated atmospheric CO_2_ concentrations. During the early season, soybeans exhibited 57% more damage from insects such as the Japanese beetle (*Popilia japonica* Newman), Potato leafhopper (*Empoasca fabae* Harris), Mexican bean beetle (*Epilachna varivestis* Mulsant), and Western corn rootworm (*Diabrotica virgifera virgifera* Le Conte) than soybean grown under ambient atmospheric conditions. This study concluded that the measured increase in simple sugars content in soybean foliage may have stimulated compensatory insect feeding [67]. Under these conditions, insect herbivores tend to consume more plant material and thereat cause more plant damage [68,69]. Increased feeding rates do not always compensate for reduced diet quality, and consumption of plants growing under elevated CO_2_ conditions could reduce the efficiency of the arthropods that feed on them [70]. Responses to CO_2_ fertilization vary depending on the type of pest feeding. Whole-cell feeders such as thrips show an increase in population size [66]. Phloem-feeding insect pests, including whiteflies and aphids, have combined responses of increased population growth rates and a decrease in population density [71]. There are inconsistent reports on the effects of elevated CO_2_ on sucking insects, although in some cases abundance and fecundity may increase [60]. Stiling and Cornelissen [72] conducted a meta-analytic study and reviewed study documentation on the indirect effects of a CO_2_ increase on life history parameters of herbivores. The results of their study showed strong responses of insect pests to increased CO_2_ compared to ambient CO_2_; (I) an increase in consumption rates of about 17%; (II) a decrease in pest abundance of about 22%; (III) an increase in development time of about 4%; and (IV) a decrease in relative growth rate of about 9% (Figure 2). Stronger effects of the increase in atmospheric CO_2_ were also found for chewers in contrast to other feeding guilds, such as sap-sucking herbivores (e.g., aphids, leafhoppers, scale insects). Thus, it has been shown that despite the numerous studies conducted to assess aphid responses to elevated atmospheric CO_2_ levels, it is still not possible to predict future response in general or to establish general rules for different aphid populations to changes in climate [73,74].

#### 2.2.3. Response of Insect Pests to Changeable Precipitation Pattern

Changes in the amount, intensity, and frequency of precipitation are very important indicators of climate change. As observed in most events, the frequency of precipitation has decreased while the intensity of precipitation has increased. This type of rainfall pattern has favoured the occurrence of droughts and floods. Insect species that overwinter in the soil are directly affected by overlapping rainfall. In short, heavy rainfall can lead to flooding and prolonged stagnation of water. This event threatens insect survival and at least affects their diapause (Figure 3). In addition, insect eggs and larvae can be washed away by heavy rains and flooding [6]. Small-bodied pests like aphids, mites, jassids, whiteflies etc. can be washed away during heavy rainfall [75] (Figure 3). Variable rainfall can have a major impact on insect populations. For example, Staley et al. [76] studied the effects of increased summer rainfall and drought on the Soil-dwelling wireworm (*Agriotes lineatus* L.) in grassland plots. Wireworms are very damaging pests of crops such as potatoes, corn, sugar beet, etc., especially when grown in grassland plots, and there are predictions that they are likely to become a much greater problem with the effects of climate change [77]. Staley et al. [76] found rapid growth of wireworm populations in the upper part of the soil as a result of increased summer rainfall events as opposed to ambient and drought conditions [78]. Herbivorous insects are affected by drought through several mechanisms; (I) dry climates may provide suitable environmental conditions for the development and growth of herbivorous insects; (II) drought-stressed plants attract some insect species. For example, when plants lose moisture through the process of transpiration, water columns in the xylem break apart or cavitate, producing an ultrasonic acoustic emission that is detected by harmful bark beetles (Scolytidae); (III) plants stressed by drought are more susceptible to insect attack because of a decrease in the production of secondary metabolites that have a defense function [79] (Figure 3).

#### 2.2.4. Expansion of Insects’ Distribution

In general, the following factors may determine the distribution of insect pests: (I) natural biogeography; (II) crop distribution; (III) agricultural practices (monocultures, irrigation, fertilizers, pesticides); (IV) climate; (V) trade; and (VI) cultural patterns [80]. Climate change will have a major impact on the geographic distribution of insect pests, and low temperatures are often more significant than high temperatures in determining their geographic distribution [81]. Numerous pest species are shifting their range because of climate change, but also due to increased international trade, which allows individuals to disperse throughout the world. In the case of agricultural insect pests, this type of dispersal shift can greatly affect agricultural production [82]. The geographic distribution and abundance of all organisms in nature is accentuated by species-specific climatic requirements that are crucial for their growth, development, reproduction, and survival. Modified temperature and precipitation patterns with the foreseeable changes in climate will determine the distribution, survival and reproduction of species in the future [43]. Due to the spread of insect pests to new areas, along with the shift in the growing areas of their host plants, farmers will face new and severe pest problems. In such cases, in addition to climatic conditions suitable for the particular crop, other factors such as soil properties and environmental structure are of great importance [83]. For pest species in general, a poleward shift in distribution limits is predicted as a response to global warming [84]. The ranges of insect pests are expected to shift to higher altitudes by 2055, with an increase in the number of generations in central Europe. In Europe, for example, the European corn borer (*Ostrinia nubilalis* Hubner) has shifted more than 1000 km northward [85]. Nevertheless, a decrease in the number of generations was predicted for southern Europe due to global warming, which would negatively affect populations of this insect pest. This implies that climate change affects various species differently [6]. Lopez-Vaamonde et al. [86] reported that 97 non-native Lepidoptera species in 20 families have become established in Europe, and 88 European Lepidoptera species in 25 families have expanded their range in Europe, with 74% of species becoming established in the last century. Parmesan et al. [87] studied 35 species of non-migratory European butterflies and concluded that the geographic ranges of 63% had shifted 35 to 240 km northward and only 3% southward in the 20th century. Increased fluctuations of warm air masses towards higher latitudes have resulted in the establishment of the Diamondback moth (*Plutella xylostella* L.) in Arctic Ocean on the Norwegian islands of Svalbard, 800 km north of its former range limit in western Russia [88]. The pink bollworm (*Pectinophora gossypiella* Saunders), a major cotton pest, is presumed to be expanding its current range from the frost-free zone in southern Arizona and California into the cotton growing areas of Central California [89]. Gutierrez et al. [90] suggest that the range of the Olive fly (*Bactrocera oleae* Rossi) in both Europe and North America will retreat southward and expand northward due to the effects of warmer summer temperatures and milder winters on adult flies. High summer temperatures currently limit the range of the *B. oleae* in the desert regions of Arizona and southern and central California, while the cold limits its range in the far north. Climate warming is predicted to further limit its occurrence in many regions of California as high summer temperatures become increasingly unfavourable. Conversely, climatic conditions along the California coast are expected to be more favourable for them to thrive. In Italy, low winter temperatures limit olive and *B. oleae* occurrence in the northern regions, but this is expected to change as formerly unfavourable regions become favourable due to global warming [90]. On the other hand, changes in frost pattern, are one of the drivers of the spread of the frost-sensitive insect pests [91]. The frequency of spring frosts decreases with increasing temperature, so longer warm periods extend the period and intensity of insect epizootics [92]. Crop growers can in theory benefit from earlier seeding, but as a consequence these plants then become available to insect pests sooner, allowing them to begin feeding earlier and cause greater damage, as well as potentially producing additional insect generations during the typical growing season [92]. In addition, rising temperatures may increase the overwintering survival of insects that were limited by low temperatures at higher elevations, leading to an expansion of their geographic range [93,94].

#### 2.2.5. Increased Overwintering Survival

Insects are poikilothermic or cold-blooded animals and therefore have a limited capacity for homeostasis in response to changes in ambient temperature. They have evolved a variety of strategies to stay alive under thermally stressful environmental conditions [95]. The most critical season for many insect pests is winter, as low temperatures can significantly increase mortality and thus reduce populations in the following season [81]. Studies have shown that global warming is most pronounced in winter at high latitudes [8]. Therefore, insects that undergo a winter diapause are likely to experience the greatest changes in their thermal environment [96]. In terms of overwintering strategies, insects are generally classified into two groups: freeze-tolerant and freeze-avoidant. The first group of insects uses a physiological adaptation strategy in the form of diapause, while the second group uses a strategy in the form of behavioural avoidance or migration [96]. Insects may enter diapause, which is an obligate or facultative, hormonally mediated state of low metabolic activity characterized by suppressed development, suspended activity, and increased resistance to adverse environmental extremes [97]. Diapause is an adaptive trait that plays an important function in the seasonal regulation of insect life cycles and is influenced by environmental factors such as temperature, photoperiod and humidity [98]. Aestivation and hibernation are two types of diapause. Aestivation allows insects to survive in environments with higher temperatures, while hibernation keeps them alive at lower temperatures [99]. In this article, we will focus only on winter diapause—hibernation.

Diapause is a fundamental requirement for overwintering success of many species in temperate and colder climates, and it confers increased cold hardiness (an organism’s ability to survive at low temperatures) in the absence of acclimation to low temperatures, which usually occurs naturally during the transition from summer to fall and winter [100]. Some insect species enter diapause during the inactive egg or pupal stages, while others do it as larvae, nymphs, or adults. When diapause occurs in the inactive stages, it is often accompanied by a sharp drop in metabolic rate that is accompanied by an increase in cold hardiness [96]. During larval diapause, which is likely more common in subterranean herbivores that are protected from low temperatures, feeding may continue and forward development may slow down rather than stop [97]. While diapause is an obligate part of the life cycle in univoltine species, it is facultative in multivoltine species and dependent on an environmental trigger such as photoperiod [97].

The adaptive significance of the seasonal response to photoperiodism is to shut down further development and reproduction by preparing metabolic activities for winter dormancy, even though current environmental conditions possibly be favourable [101]. Moreover, considering the complex roles insects play in the ecosystem, many other processes are synchronous with their diapause programme, such as plant consumption, pollination, or interspecies interactions. Consequently, a single disruption of diapause as a result of anthropogenic climate change can have profound effects on the stability of the entire ecosystem. Therefore, when discussing the effects of climate change, it is important to consider the effects of climate warming on all three phases of diapause, namely diapause initiation, diapause, diapause termination, and post-diapause quiescence [96]. For many insect species, it is likely that higher temperatures during the photoperiodic induction of diapause (usually in autumn) reduce the frequency and duration of diapause [96]. In the European bluebottle fly (*Calliphora vicina* Robineau-Desvoidy), for example, adult flies reared at 20 °C produce a smaller number of diapausing offspring. Diapause is also shorter than in flies reared at 15 °C [96,102]. In cases where diapause is obligate for successful overwintering, higher temperatures are required to allow development to the next diapausing generation before severe winter conditions begin. In addition, for many temperate insect species, delaying diapause poses the risk of encountering cold stress outside diapause or before cold tolerance mechanisms are established [101].

This is believed to determine the current northern range of the Green stink bug (*Nezara viridula* L.) in Japan. Only diapausing adults of *N. viridula* are capable of overwintering, and winter conditions (timing of diapause induction) at the northern border begin when stink bugs reach only the nymphal stages. The entire population is, therefore, doomed to decline [103]. Further south, however, there is a sufficiently long growing season for this generation to reach the adult stage before winter, and at these sites *N. viridula* has displaced the most recently dominant pest specie, the Oriental green stink bug (*Nezara antennata* Scott) [96,104]. The duration of diapause can be influenced by many factors: accumulated chilling, humidity, food, and photoperiod [97]. However, for many species, the general principle is that the duration of diapause is shorter at higher temperatures. For example, the flesh fly (*Sarcophaga crassipalpis* Macquart), a common laboratory insect used to study diapause processes, remains in diapause for 118 days at 17 °C, 70 days at 25 °C, and 57 days at 28 °C [96,105]. This is because warmer winter temperatures increase the metabolic rate of diapause, resulting in a shorter diapause. A comparison of metabolic rates and diapause duration under these different conditions suggests that diapause ends when energy reserves reach a critical point. When metabolic rate is high, energy reserves are depleted quickly, and when metabolic rate is low, this set point is reached much later, resulting in a longer diapause [105].

Good synchronization with the environment and host plant means that insect herbivores are well adapted to their habitats [106]. However, climate warming can disrupt the metabolic balance during diapause, which can significantly affect the timing of emergence, so any change in spring emergence could lead to a loss of synchrony with the environment or host plant [96,106]. For example, many insects rely on synchrony between the timing of bud burst (or flowering) and emergence of feeding stages. It is quite conceivable that under current predictions of climate change, synchrony between trophic levels could become uncoupled as a consequence of subtle environmental differences in the phenology of individual species [107]. One of the best-studied examples of this is the Winter moth (*Operophtera brumata* L.), in which egg hatching is markedly advanced compared to bud break on its host plant, the Pedunculate oak (*Quercus robur* L.). It is unlikely that a 2 °C increase in temperature will dramatically alter the timing of bud burst, but the timing of larval hatching is likely to be significantly advanced, possibly leading to larval hatching before bud burst, which is dangerous for the moth and could reduce this specific pest problem [108].

It appears that univoltine temperate species respond differently to warmer winter conditions, making it difficult to predict the precise effects of climate change on overwintering insect species [109]. Non-diapausing, frost-sensitive species and those that can overwinter in their active stages appear to have increased survival rates under warmer winter conditions. These insect pests are expected to build up their populations and expand their geographic ranges to higher altitudes as average temperatures there increase [49]. Extremely low winter temperatures increases winter mortality, which is considered a key factor in the dynamics of many temperate insects, especially those that do not go into diapause but are active throughout the winter when temperature permits [108]. Warmer winters or a reduction in the frequency of extreme cold periods may, therefore, improve the survival of such species, as they are not exposed to low lethal temperature extremes [96]. However, insects exhibit a variety of strategies in relation to the threat of lethal low temperatures and these will partly determine the impact of warmer winter conditions [110].

Increased survival during overwintering period could lead to an increase in overwintering population and therefore to a greater abundance of insects on plants during the warmer period of a year. Consequently, global warming would increase the build-up of insect populations, early infestations and resultant crop damage from insect pests [111,112]. For example, increases in temperature have resulted in range expansion and increased overwinter survival of the Corn earworm (*Helicoverpa zea* Boddie) and the Cotton bollworm (*Helicoverpa armigera* Hubner). Consequently, this appears to be a significant threat to yield loss and a major challenge for pest management in corn, a fundamental food crop in the United States [43,113].

The flight phenology of aphids can be an accurate biological indicator of climate warming [114]. Many authors have shown that an increase in temperature promotes the survival of overwintering anholocyclic aphid species in the United Kingdom and in some cases brings forward their flight onset by up to one month. Such changes caused by climate warming will increase aphid outbreaks and cause earlier spring migrations, giving populations a better chance to build up to damaging levels in the subsequent growing season with a prolonged virus infection period [114,115,116,117]. Horticultural pests of plants grown in and restricted to greenhouses will have more opportunities to survive outdoors as average temperatures increase. For instance, warmer winter conditions are likely to increase the probability of the invasive South American leaf miner (*Liriomyza huidobrensis* Blanchard) overwintering outside greenhouses in the United Kingdom [114,118].

#### 2.2.6. Increased Number of Generations

As mentioned earlier, temperature is the most important environmental factor for insects, affecting mainly their phenology. The ambient energy hypothesis suggests that growth and reproduction are greater at high temperatures. Therefore, higher temperatures or global warming leads to higher population sizes, which in turn can lead to a higher number of species in dynamic equilibrium [119,120]. Under a global warming scenario this makes it possible to accelerate reproductive rates within a certain preferred range, leading to an increase in the number of generations of many insect species and to more crop damage [121]. One of the many species traits and climate variables that have been used to link climate change to phenological shifts is thermal development tolerance, which can be measured using growing degree days (GDD).

GDD is a measure of heat accumulation calculated annually by accumulating the daily total sum of degrees between a minimum and maximum temperature threshold (Dmin and Dmax). GDD has long been used to predict plant and insect phenology in agriculture [122]. Future temperature increases will affect univoltine and multivoltine temperate species in different ways and to different extents. For multivoltine insects, such as aphids and some lepidopteran species, such as the large cabbage white butterfly (*Pieris brassicae* L.), higher temperatures, all other parameters being equal, should allow for faster development times that predictably allow for additional generations within a year [49,123]. Species with annual life cycles generally develop more rapidly than those with longer life cycles [49]. Using several models, it has been extrapolated that a 2 °C increase in temperature could result in one to five additional life cycles per year [121]. The most significant examples in this regard are aphids, which can be expected to produce four to five additional generations per year due to their low developmental threshold and short generation time. Aphids may, therefore, be particularly sensitive indicators of temperature changes [120]. Higher temperatures during their development have the beneficial effect of shortening the time in the larval and nymphal stages (when they are highly threatened by predators) [124], and allowing species to become adults earlier [120].

Expected responses of insects to a rise in temperature include an advance in the timing of adult emergence and an increase in flight duration [120]. One explanation for the changes in voltinism is the earlier onset of the flight period, which could allow for the production of an additional generation [125]. Since the insects fly earlier in the growing season, the individuals of the first generation could reproduce earlier. In addition, due to higher temperatures, faster larval development and growth occurs, so more individuals of the subsequent generation could develop when photoperiod and temperature conditions are still favourable, allowing them to develop directly in the same season rather than diapausing as larvae [125]. The timing of adult emergence can be documented with pheromone-, suction-, or light- traps. Long-term data analyses on insect phenology show that the timing of emergence of insect pests changes under climate change [75]. Analysis of suction trap data showed that the spring flight of the Potato aphid (*Myzus persicae* Sulzer) began two weeks earlier for every 1 °C increase in mean temperature in January and February [6]. Depending on the temperatures during winter and the duration of exposure, the relative abundance of populations after winter ranges from very low (cold winter) to very high (mild winter) [126]. A 50-year report of the timing of the first migrating individuals of *M. persicae* caught in a suction trap each year (from a study by Rothamsted Research, Harpenden, UK) showed a strong correlation with winter mean temperatures in January and February [96]. Members of the order Lepidoptera are another good example of phenological changes. Such changes in butterflies have been reported in the UK, where 26 of 35 observed species have advanced their first appearance [120,127]. In Spain, the first appearance of 17 species has shifted by 1–7 weeks in only 15 years [120,128]. Early emergence increased voltinism in the European grapevine moth (*Lobesia botrana* Denis and Schiff.) in Spain. This pest is usually trivoltine in Mediterranean latitudes, but with a tendency to emerge early in spring, it sometimes has a fourth additional flight, possibly due to global warming [129]. Since the 1980s, the number of generations per year has increased in many central European Lepidoptera species, with some univoltine or bivoltine species transitioning to bivoltine or multivoltine life cycles [125]. Partially bivoltine or multivoltine species are expected to experience an increase in abundance of second or subsequent generations [125,130].

Given the wide diversity of insect pests, it seems impossible to describe the precise effects of climate change for each species, the environmental conditions and the ecosystems in which they interact [49].

However, accurate quantification of the relationship between climate change and insect traits, such as changes in phenology and voltinims for a key insect pest species, could provide a conceptual framework for how these specific changes might manifest in other insect species [131]. The documented changes in voltinism confirm the high adaptability of insects to environmental change, which is why they are among the organisms that respond to global warming [121].

#### 2.2.7. Increased Risk of Invasive Alien Insect Species

Invasive alien species (IAS) are defined as taxa that are introduced either intentionally (e.g., food, crops, ornamentals, pets, livestock) or unintentionally due to human activities outside their natural habitat [132]. Invasive insects are usually agricultural, stored-product, forestry, household or structural pests and can often be vectors of various diseases or parasites [133]. The spread of species to regions outside their original range has accelerated exponentially over the last millennium due to international travel, the global trading system and agriculture [134]. The Convention on Biological Diversity [135] describes invasive alien species as the greatest threat to global biodiversity with high costs to agriculture, forestry and aquatic ecosystems [6]. It is commonly assumed that only a small proportion of introduced IAS become established and only a small proportion of these species spread and become economic pests. This is often referred to as the “rule of 10,” according to which approximately 1 in 10 introduced species escape into the environment, 1 in 10 of these introduced species become established in the environment, and 1 in 10 of these established species become economic pests [136].

For invasive insect pest species, many authors in recent studies predict expanded geographic range and increased population densities and voltinism under predicted climate change scenarios [49,126,137,138], which could soon lead to potentially severe consequences for sustainable agricultural production [139]. However, it is important to state that climate change is not the predominant driver of biological invasion. To become invasive, alien insects must successfully arrive in a new habitat, survive the given conditions, and thrive. Climate change could positively or negatively influence the components of this invasive pathway. Climate, in combination with landscape features, sets the limits for the dispersal of such species and determines the seasonal conditions for their development, growth and survival in a new habitat [140]. These habitats may have been previously unsuitable, and dispersal to suitable, distant habitats may have been blocked by a geographic barrier, such as mountain ranges or the sea [141]. All biological systems have thermal limits, so temperature increase will have a huge impact on ecosystems and the species that live in them.

The extent of the responses of most native and non-native insect species to global warming is still unknown, and certainly the new warmer conditions would not be beneficial to all of them [140]. The process of insect invasion involves a chain of events that include the transport, introduction, establishment, and dispersal of invasive alien insects [134]. Once a new species arrives in a new habitat, the other stages of the invasion process could be positively or negatively influenced by existing climate and climate change [142]. Climate change can directly affect the transport and introduction of invasive insects. Extreme climate events (e.g., storms, high winds, hurricanes, currents, and swells) could shift pests to new geographic areas where they may find environmental conditions favourable for establishment [143]. For example, the Cactus moth (*Cactoblastis cactorum* Berg), was blown from the Caribbean islands to Mexico during the 2005 hurricane season, where it posed a significant ecological and economic threat to more than 104 prickly pear species (*Opuntia* Mill), 38 of which are endemic [144,145]. Some insect species are more prone to introduction and dispersal to new geographical regions than others, and some pathways favour the introduction of some alien insect species [146]. The number of insect individuals arriving is referred to as propagule pressure [141], also known as “introduction effort” [147].

Propagule pressure is a function of the frequency and number of individuals invading a new habitat [133]. In general, the more individuals introduced into an area, the greater the chance that they will successfully establish [147]. One or more propagules of a species must first enter a transport pathway, then survive the transport journey, followed by a successful exit from the transport vector, and final establishment of an initial population that may or may not spread and become invasive [148]. Propagule pressure is related to the extent of plant trade, the likelihood that alien insects are transported on these plants, and the probability that they pass through border controls undetected in plant commodities [149]. One of the most recent examples of such an introduction pathway is the case of the invasion of the highly polyphagous and harmful invasive insect, the Spotted wing drosophila (*Drosophila suzukii* Matsamura), in North and South America and Europe. The pathway of introduction is thought to be trade in fresh fruit, with initial propagules occurring undetected in the egg or larval stage in large quantities of fresh fruit traded via South East Asia [150,151]. The spread of invasive pest species due to climate change is in fact, slow. Parmesan and Yohe [19] found that insect species are spreading at an average rate of 6.1 km per decade due to climate change. This is happening due to the increase in temperature in these areas and is causing insects to survive where they could not previously thrive [92].

Invasive species usually have a wider range of tolerance or bioclimatic range than native insects, allowing alien insects to find a wider range of suitable habitats [137]. Insect species are known to be highly sensitive to climate change. Sensitivity arises from the fact that most of their physiological processes are temperature-dependent [152]. Plasticity is a driving force behind the spread of many invasive species. Because plasticity is a trait of the individual, it is often touted as a responsive mechanism that allows organisms to adapt to new environmental conditions in the rapidly changing world (also referred to as “plastic rescue”) [153,154].

Adaptations can take the form of phenotypic, behavioural, developmental, or physiological traits. Physiological or behavioural plasticity may result from differences in environmental conditions (e.g., temperature, humidity, photoperiod), available diet, or pressure from predators or competitors [155,156]. Behavioural responses can be adaptive and improve fitness, such as finding host plant species when invading new environments. One of the plastic responses of foraging insects to new environments is to change or expand their food choices. For some species, such as *D. suzukii*, which shows extreme plasticity in its diet choice with more than 30 plant species, diet breath is probably the most important trait responsible for its invasion success [157]. The evolution of many traits involves components of different mechanisms, such as plastic responses to photoperiod in relation to climate change [153]. Snell-Rood et al. [153] predict that general mechanisms that evolve through selective processes within an individual are very likely to lead to survival in new environments, especially when conditions exceed the typical range of the native environment in extreme ways, such as large temperature shifts. In ectotherms such as insects, thermal adaptation may occur, for example, through behavioural traits that control energy metabolism [158].

#### 2.2.8. Reduced Effectiveness of Biological Control Agents—Natural Enemies

Climate change is likely to have severe impacts on the abundance, distribution, and seasonal timing of pests and their natural enemies, which will alter the degree of success of biological control programs [69]. Phytophagous insect species are naturally controlled by top-down (natural enemies) and bottom-up (host plant availability and quality) mechanisms. These natural mechanisms interact to influence insect population dynamics, performance, and behaviour [159]. In agricultural, forestry and other ecosystems, phytophagous insects can be considered as cornerstones of the tri-trophic host plant–insect pest–natural enemy relationship [160]. The effects of climate change on interactions between insect pests and natural enemies, whether natural enemies are intentionally introduced to new regions or whether they are native and biological control is supported by conservation measures, are modulated by direct effects on the metabolism and physiology of the organisms involved, the responses of those organisms, and subsequent tri-trophic interactions. These interactions are affected by climate change in a variety of ways. Temperature changes can affect the biology of each component species of a system differently, destabilizing their population dynamics [60] and causing temporal desynchronization. Natural enemies, which are the third trophic level, are expected to be significantly affected by climate change [161]. If trophically connected species respond variously to climate change, the trophic interaction between them could be perturbed, resulting in decoupling of the synchronized dynamics between insect pests and their natural enemies and potentially negatively affecting the performance of biological control [162].

Aphids are among the insect pests that are controlled by many natural enemy species, such as parasitic wasps, which lay their eggs in the bodies of aphids, and predatory species, such as ladybirds. All of these species are affected by the effects of global warming and could respond differently to temperature changes [73,107]. Hance et al. [60] reported that if a natural enemy starts to develop at a slightly lower temperature than the prey (e.g., aphid) and develops faster than the prey when the temperature rises, a too early and warm spring leads to its early emergence and a high probability of death from lack of prey. If this phenomenon is repeated over several years, it may lead to the extinction of the natural enemy. Evans et al. [163] showed that a rise in temperature disrupted the biological control of the Cereal leaf beetle (*Oulema melanopus* L.). In this trophic system, the development of *O. melanopus* was more affected by warming than that of the natural enemy, resulting in a phenological shift between enemy and prey and a weakening of biological control.

Crop distribution ranges are predicted to shift due to climate change. As an outcome, herbivores may track changes in crop distribution and migrate to areas where they may or may not be tracked by their predators or parasitoids, resulting in spatial desynchronization [73]. The final outcome depends partially on the competence of corresponding natural enemy species to expand their geographic range or on the possibility of new natural enemy populations that could control the pest in its new habitat [69]. In the absence of these conditions, herbivores may be able to escape predation and build large populations in their new habitat [164]. The potential for natural enemies to pursue their hosts depends primarily on their environmental tolerance relative to their herbivorous hosts, as well as their movement rates [69]. Gilman et al. [165] suggested that natural enemies that are specialists are more likely to be affected by climate change than generalists because they are less able to adapt to spatial desynchronization with their host communities. In such a case, biological control in food webs composed of many generalists might be more resilient to climate change [164].

Elevated CO_2_ concentration, altered precipitation patterns, and temperature increase modify plant phenology and productivity, which in turn affect the growth and abundance of herbivore populations (host insects) and indirectly influence the supply of prey and hosts available for predation or parasitism [64,69]. Thomson et al. [69] also found that plants grown under elevated CO_2_, temperature extremes, and reduced precipitation provide diverse nutritional resources for herbivores, indirectly affecting the fitness of parasitoids and predators that feed on these herbivorous hosts. Bezemer et al. [66] studied the effect of temperature increase and elevated CO_2_ concentration on the synchronized population of a tritrophic system consisting of the host plant annual bluegrass (*Poa annua* L.), the pest green peach aphid (*M. persicae*) and the parasitoid wasp (*Aphidius matricariae* Haliday). They showed that aphid populations built up under both elevated temperature and elevated CO_2_. There was no information on an effect of elevated CO_2_ on parasitism success, but parasitism increased in correlation with elevated temperature. Another study examining the efficiency of the parasitoid wasp (*Aphidius picipes* Nees) feeding on the English grain aphid (*Sitobion avenae* F.) showed that parasitism increased in correlation with elevated CO_2_, but the same elevated CO_2_ level resulted in lower fecundity of the wasps [166].

Therefore, the overall competence of a given species under elevated CO_2_ concentrations is positively or negatively affected depending on its life history traits [166]. Dyer et al. [167] found that elevated temperature and CO_2_ reduced the nutritional properties of alfalfa plants, (*Medicago sativa* L.), which are the host plants of the Beet armyworm (*Spodoptera exigua* Hubner). This type of reduced nutritional quality of the host plants resulted in a shortened development time of the larvae of *S. exigua*. At the same time, larvae of its natural enemy, the parasitic wasp (*Cotesia marginiventris* Cresson) were unable to fully develop, leading to the extinction of the local population of *C. marginiventris*. Few studies have addressed how increased CO_2_ concentration affects predator efficiency. The family of ladybirds (Coccinelidae) is the largest insect group of predatory natural enemies. Chen et al. [166,168] investigated the food preference of the Asian ladybird (*Harmonia axyridis* Pallas) in food choice experiments. They showed that *H. axyridis* preferentially preyed on aphids under elevated CO_2_ concentration compared to ambient CO_2_ concentration. Despite this preference, predation performance was not affected by high CO_2_ concentrations. Eventually, the time required for larval development of *H. axydiris* was significantly shorter or remained unchanged under altered CO_2_ conditions [166,168].

Ultimately, climate change and global warming affect higher trophic levels directly, by altering the behaviour of natural enemies, or indirectly, by altering physiological traits in host plants and behavioural traits in herbivorous insects. Given all these facts, it is important to assess the trophic system as a whole. A challenge for the future is to develop models based on knowledge of phenological processes obtained through long-term monitoring of herbivores, their associated natural enemies and host plants, and their response to current climate and climate change [69].

#### 2.2.9. Increased Incidence of Plant Diseases Transmitted by Insect Vectors

Insects are important vectors that transmit many plant diseases such as viruses, phytoplasmas and bacteria [169]. Viruses are a major cause of many plant diseases in global food production. The estimated economic loss from these diseases exceeds $30 billion per year [170]. Outside their vector or host insect, viruses are immovable and therefore heavily dependent on their vectors for transmission and spread. Some viruses and vectors are host generalists and others are specialists with a specific mode of transmission. Vectors can vary in their transmission efficiency, so the persistence, spread and prevalence of viruses depend on the particular vectors, their host plant and the climatic conditions in which they thrive [171,172]. Climate change may have a major impact on the epidemiology of plant viruses [173]. Most viruses of agricultural crop species are messenger RNA viruses and single-stranded DNA viruses. Their main host-to-host transmission strategy is the use of insect vectors with mouthparts for piercing and sucking [174]. In the previous sections, we have described the effects of climate change on various insect pests, some of which act as vectors of viruses. As climate directly affects insect physiology, phenology, etc., it could indirectly affect the viruses they transmit. This influence could have positive, negative or neutral consequences for the emergence and development of viral diseases in crop production [172].

Global warming may favour the occurrence of insect-transmitted plant diseases due to geographic expansion and increases in populations of insect vectors [175,176]. The main order of insects that transmit plant viruses are the sap-feeding Hemiptera. Within this order, the families of aphids (Aphididae), leafhoppers (Cicadellidae) and whiteflies (Aleyrodidae) are the major vectors of viral diseases [177]. Among these, aphids are the largest group of vectors, transmitting more than 275 virus species, and the majority of aphid species are capable of transmitting some plant viruses. Aphids are crucial virus vectors in temperate zones of the world, while whiteflies are restricted to warmer areas and thrive in temperate regions in crops grown under greenhouse conditions [174]. The short development time and high reproductive capacity of aphids and whiteflies make them particularly sensitive to responses to climate change [60]. The migration potential and long-distance dispersal of virus vectors could also be affected by climate change. Aphids can travel long distances when they encounter favourable thermal conditions that launch them upward, where atmospheric air movements expose them to horizontal translocation [178]. This long-distance transport has been linked with severe viral epidemics caused by aphids transported by extremely persistent low-pressure winds from the Great Plains of North America in the south to corn-growing areas in Minnesota [179].

It has also been reported that an increase in temperature in Northern Europe, especially at the beginning of the growing season, increases the rate of viral diseases in potato due to earlier colonisation by aphids, the main vectors of potato viruses [43,180]. The severity of viral diseases is highly dependent on the timing of infection and the amount of inoculum. The amount of viral inoculum is influenced by the overwintering of its insect vectors and their (alternative) host plants [181]. Aphids are expected to have higher survival rates in milder winters, and higher spring/summer temperatures increase their development and reproduction rates. The final outcome is a higher incidence of viral disease transmission and spread [182].

Barley yellow dwarf virus (BYDV) causes a very damaging disease in the Poaceae family and is transmitted by various aphid vectors. In Central Europe, the temperature minimum for migration of the Bird cherry oat aphid (*Rhopalosiphum padi* L.), the main vector of BYDV, is 8 °C, based on long-term monitoring. In addition, population build-up in summer is determined by temperatures in autumn, and population build-up in autumn is dependent on precipitation patterns and extremely low temperatures in winter [183]. Warmer conditions in autumn and winter in central and northern Europe increase vector persistence and thus the risk of virus transmission in winter crops such as winter barley and winter wheat [184]. In summer, warm temperatures and low rainfall reduce host availability, which poses various challenges for viruses and their insect vectors. Temperatures above 36 °C in warmest summer months result in decreased survival of aphids, reducing the spread of BYDV [185].

Among whiteflies, the Greenhouse whitefly (*Trialeurodes vaporariorum* Westwood) and the Silver leaf whitefly (*Bemisia tabaci* Gennadius) are most important virus vectors. Moderate precipitation and high temperatures are generally favourable for *B. tabaci* and lead to population increases [186]. Environments with dry and hot climates with installed irrigation systems provide favourable conditions for *B. tabaci*. Considering their short generation time, large populations can develop in summer. The same conditions could lead to an increase in the rate of evolution of the virus, resulting in more efficient strains with broader host range, greater transmission efficiency, and larger virus reservoirs in crops. Extreme winds and increased cyclonic activity in the tropics, as predicted by climate change scenarios, could promote the spread of *B. tabaci*. Drought could decrease its survival rate and disrupt its development, as well as restrain population size and dispersal [71]. Based on climate models, under four different climate scenarios that include data on humidity, temperature, and atmospheric CO_2_ levels, it is predicted that many more geographical regions worldwide will be suitable for outdoor tomato production. These regions may also become suitable for the establishment of *B. tabaci* populations and thus for the increased incidence of the highly damaging pathogen of tomato—tomato yellow leaf curl virus (TYLCV) [187].

Grapevine yellows are grapevine diseases associated with phytoplasmas. They show notable differences in epidemiology due to the different life histories of their associated insect vectors [188]. One of the most important grapevine diseases in Europe is Flavescence dorée [189], and its main vector is the American grapevine leafhopper (*Scaphoideus titanus* Ball) [190]. As average temperatures increase during the growing season, *S. titanus* is expanding its range northward [191]. While short summers are considered a barrier to the northern spread of *S. titanus* due to the insect’s inability to reach its full life cycle [192,193], climate change with longer and hotter summers should favour the spread of *S. titanus* in northern vineyards such as in Germany by extending the favourable development period [191]. Currently, *S. titanus* is widely spread in many vine growing areas across the Europe. Mirutenko et al. [193] reported the occurrence of *S. titanus* in Ukraine, which is currently its northern limit of distribution in Europe. However, climate warming at the southern limit of its current range could lead to insect declines or extinctions of small populations in areas such as southern Italy [192].

With climate change, an increase in newly introduced insect-transmitted plant diseases is expected. Therefore, it is of great importance to have diagnostic tools and appropriate personnel to detect new pathogens.

## 3. Adaptation and Mitigation Strategies for Pest Management in a Changing Climate

Climate change adaptation can be viewed as an ongoing process of implementing existing risk management strategies and reducing the potential risk from climate change impacts [194]. Climate change is widely expected to make pest infestations more unpredictable and increase their geographic range. Coupled with the uncertainty of how climate change will directly affect crop yields, the interactions between insects and plants in ecosystems remain unclear [78]. The adaptive capacity of agricultural production systems will depend on several biological, economic, and sociological factors. The ability of local communities to adapt their pest management practices will depend on their physical, social and financial resources [71]. With climate change and the acceleration of global trade, uncertainties and frequency of occurrence of existing and new pests will increase. Increasing the ability to adapt rapidly to disturbances and climatic changes will therefore become all the more important [195]. Potential adaptation strategies have been identified to reduce the risks of spreading new pests and diseases, and to mitigate the negative impacts of existing pests. The most commonly mentioned strategies are modified integrated pest management (IPM) practices, monitoring climate and insect pest populations and the use of modeling predictions tools [92] (Figure 4).

### 3.1. Modified Integrated Pest Management (IPM) Practices

By definition, IPM refers to harmful species of phytophagous animals (mainly insects and mites), pathogens and weeds. In the context of sustainable agriculture, the emphasis in plant protection is on preventive or indirect measures, which must be fully exploited before control or direct measures are applied. Decisions on the need for control measures must be based on the most modern tools, such as forecasting methods and scientifically validated thresholds. Direct pest control tools are a last resort when economically intolerable losses cannot be prevented by indirect measures [196].

FAO recommends a dual strategy based on action at global and regional levels and, above all, significant investment in improving existing early detection and control systems. This requires the development of new agricultural practices, the introduction of new crops species, and the application of the principles of integrated pest management to contain the spread [197].

Mainly, growers and researchers design IPM strategies to minimize negative impacts on the environment while maximizing crop yields and economic returns [198]. Many authors have discussed the problem of pest management in a novel environment with a changing climate and the need to reconsider existing preventive agricultural practises and IPM strategies to improve heterogeneous agroecosystems that are resilient enough to tolerate weather variability [195]. However, in recent years, it has been predicted that researchers and growers will need to change many of these carefully constructed IPM tactics to respond to the important impacts of global warming [195].

Many IPM programs have focused on decisions based on extensive knowledge of how many insect pests can be tolerated before economic yield losses occur, also known as economic or intervention thresholds. IPM has historically evolved in the pest management field where the use of established thresholds has yielded good results. Although intervention thresholds play an important role in IPM, they are not always relevant, sufficient, or possible. When decision support systems are not available or appropriate, the use of thresholds is neglected [195]. Understanding how the environment affects plant and pest development is critical, and understanding their interactions with the environment allows crop advisors to respond to climate change. Environmental factors such as drought stress affect crop protection recommendations. When a crop is under drought stress, it is less able to cope with the additional stress caused by herbivorous insects, which can easily lower the economic threshold [199]. Due to the faster development of insects at higher temperatures, populations develop faster and crop damage occurs earlier than currently expected. Therefore, treatment thresholds based on the number of insects per plant must be lowered to prevent unacceptable yield losses [200].

Modified cropping practices and adaptive management strategies are needed to reduce the impact of agricultural pests on crops in a changing climate. These may include: (I) planting different crop varieties; (II) planting at different times of the year to minimize exposure to pest outbreaks; and (III) increasing biodiversity at field margins to increase the number of natural enemies [69,201].

The use of pheromones and allelochemicals is an important method by which insects sense their environment. They play a substantial role in various IPM techniques such as biological control, mating disruption, push-pull strategies, monitoring and trapping [202]. As the climate warms and microclimates become more variable, the use of pheromones and allelochemicals in their current form is expected to become less effective and may require a synergist or other adjuvant to reduce their volatility under high temperature conditions [201]. In addition, some biopesticides based on enthomopathogenic viruses, fungi, bacteria, and nematodes are extremely susceptible to environmental changes. An increase in temperature and a decrease in relative humidity may cause some of these management techniques to be less effective, and a similar result is expected for synthetic insecticides [203]. In this context, the focus should be on the development of new pest management strategies and possible new formulations of insecticides as well as attractants and repellents. For example, Wenda-Piesik et al. [204] in their study, investigated the behavioural response of confused flour beetle (*Tribolium confusum* Du Val) to different concentration of environmentally friendly volatile organic compounds (VOC) in terms of their repellent and attractive properties. As a result, they confirmed that highest concentration of applied VOC repelled individuals of given species significantly. This research can serve as a basis for the development of new sustainable and environmentally friendly pest control agents.

There is an urgent need to better understand the effects of global warming on the performance of many synthetic insecticides, their persistence in nature, and also the development of resistance to certain insecticides in pest populations [205]. Therefore, it seems necessary to consider the use of efficient biological control agents or the introduction of insect pest-resistant crop varieties obtained through conventional genetic breeding or genetic engineering [197].

### 3.2. Monitoring Abundance and Distribution

One of the most important prerequisites for determining whether climate change is altering the population dynamics of insect pest species is access to long-term data [206]. Without these important baseline data, it is extremely difficult to fully assess changes in pest populations under changing climate regimes and also to predict future population dynamics [201]. Long-term monitoring of pest populations and behaviour, particularly in climate change-sensitive regions, may provide some of the first clues to biological responses to climate change [207]. Changes in the dynamics of vectors, diseases and host populations at the local level need to be monitored, as do changes in their geographical distribution. New invasive species are being introduced in many parts of the world, aided by climate change. Effective monitoring and management systems are needed to prevent invasive species from becoming an economic pest in new geographic regions [207,208]. Therefore, adaptive responses in both pest management and biosecurity will be required.

Currently available pest management strategies such as detection, prediction, physical control, chemical control, and biological control could be intensified to control pests in response to climate change [207]. Due to the transboundary nature of many insect pests, a global management approach is needed for monitoring and risk assessment to be effective. A global system for sharing information between regions, including important information on insects, invasive alien species, diseases, and ecological conditions, including weather data, is needed. Therefore, it is important to improve cooperation between countries and regions, including national, regional, and global organizations [209]. Entry point monitoring and rapid eradication, as exemplified by the US Department of Agriculture’s (USDA) Early Warning and Rapid Response Program and the European and Mediterranean Plant Protection Organization’s (EPPO) Early Warning and Information System for IAS, will continue to be important when addressing invasive species [139,210].In addition, by monitoring climate and pests in combination with climate and pest risk prediction information, farmers can preemptively adopt certain pest prevention practices to reduce the occurrence and increase of expected pest problems [207].

### 3.3. Climate Forecasting and Model Development

It is impossible to design a priori climate change adaptation strategies for specific national or global climate change scenarios because of the heterogeneity of changes in average temperature and other climate parameters around the world. Adaptation strategies to climate change must be one of the components of an integrated strategy that takes into account all aspects of agricultural production.

Pest management strategies must tolerate regional climate change and its uncertainties. Some of the available options include sensitivity analyses and combined results obtained by using projected climate change scenarios with sensitivity analyses for a given area over a wide range of variable values. This strategy could become a useful tool in informing pest management personnel when designing adaptation measures for pest management under new environmental conditions [71].

Climate models combined with the environmental requirements of a particular pest species (envelope) can be an effective tool for projecting the possible range of changes on a global scale. Modelling the pest risk together with the responses of its plant hosts to climate change can therefore increase the ability to predict the outcome of an insect infestation [92]. The potential distribution of insect pest species is primarily estimated by ecological niche models (ENMs). They can be divided into two groups: correlative models and mechanistic models. Correlative models use correlated values of environmental variables and records of occurrence to make predictions about potentially adequate areas for the particular species. The most commonly used correlative models are MaxEnt, Bioclim, Random Forest, etc. [211]. As cited by Evans et al. [212], correlative species distribution modelling is the most commonly used approach for predicting the impacts of climate change on biodiversity and has become a cornerstone of climate change policy [213]. Correlative modelling is a widely used tool for projecting future changes in the geographic distribution of species, assessing extinction rates, and setting priorities for biodiversity conservation [214]. These models identify statistical relationships between the current geographic distributions of a given species and climate variables, which are then implemented to projections of climate change to suggest climatically suitable habitats for that species in the future [215]. The final output of correlative models is often presented in the form of maps showing future climatically adequate regions for a given species, the total area of which can then be compared with current geographic ranges to estimate the future risk of their introduction and establishment [212].

Mechanistic models are predictive tools that use the values of environmental variables of a given area in combination with knowledge about the environmental tolerances of a given species [211].

Mechanistic species distribution models differ from correlative models in that they examine how the environment constrains physiological performance in a given region. Future species distributions are then predicted through a process of elimination, whereby regions that constrain physiological performance to the extent that they affect the ability to survive, grow, or reproduce are excluded from the final distribution [216]. It has also been argued that mechanistic models are the preferred approach to most management questions because they are able to extrapolate beyond known conditions and isolate traits that determine biogeography [217]. CLIMEX is an example of a semi-mechanistic modelling software tool that uses the physiological and behavioural parameters of species and the values of climate variables to make predictions about suitable habitats or regions for specific species [218]. In addition, comprehensive analysis of climate and historical weather records, together with development of the models described above, will facilitate prediction of pest risks. This could be reflected in the development of proactive strategies for pest prevention and control strategies in a changing climate [219].

## 4. Conclusions

Although there are still many unknowns related to climate change, it is widely accepted that it greatly affects the cultivation of agricultural plants as well as the insect pests associated with them. Some of the uncertainties regarding different aspects of climate change that are relevant to insect pests include small-scale climate variability such as temperature increase, increase in atmospheric CO_2_, changing precipitation patterns, relative humidity, and other factors. Given the enormous heterogeneity of insect species, their host plants and global climate variability, mixed responses of insect species to global warming are expected in different parts of the world. The effects of climate change on insects are complex, as climate change favours some insects and inhibits others, while impacting their distribution, diversity, abundance, development, growth and phenology. Additionally, it is generally expected that there will be an overall increase in the number of pest outbreaks involving a broader range of insect pests. Insects would likely expand their geographic distribution (especially northward). Due to increased overwintering survival rate and the ability to develop more generations, the abundance of some pests will increase. Invasive pest species will likely establish more readily in new areas and there will be more insect-transmitted plant diseases. Another negative consequence that could occur as a result of climate change is the reduced effectiveness of biological control agents—natural enemies—and this could be a major problem in future pest management programs. If climate change factors lead to favourable conditions for pest infestation and crop damage, then we face a high risk of significant economic losses and a challenge to human food security. A proactive and scientific approach will be required to deal with this problem. Therefore, there is a great need for planning and formulating adaptation and mitigation strategies in the form of modified IPM tactics, climate and pest monitoring, and the use of modelling tools.

## Figures and Tables

**Figure 1 insects-12-00440-f001:**
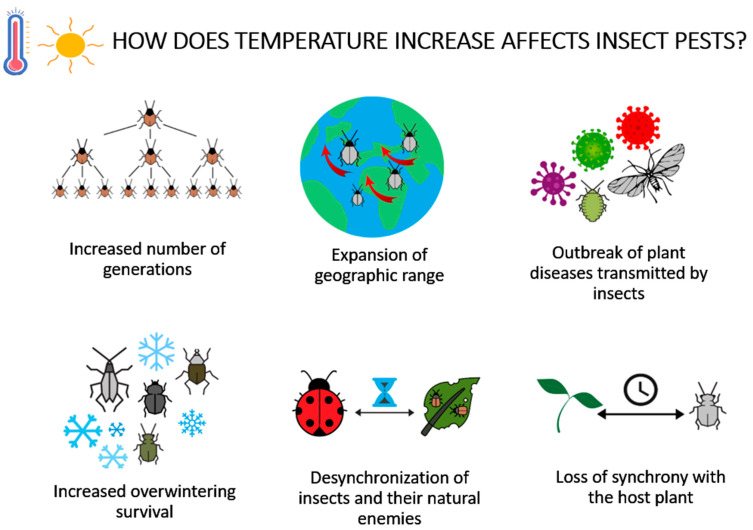
Effects of temperature rise on agricultural insect pests.

**Figure 2 insects-12-00440-f002:**
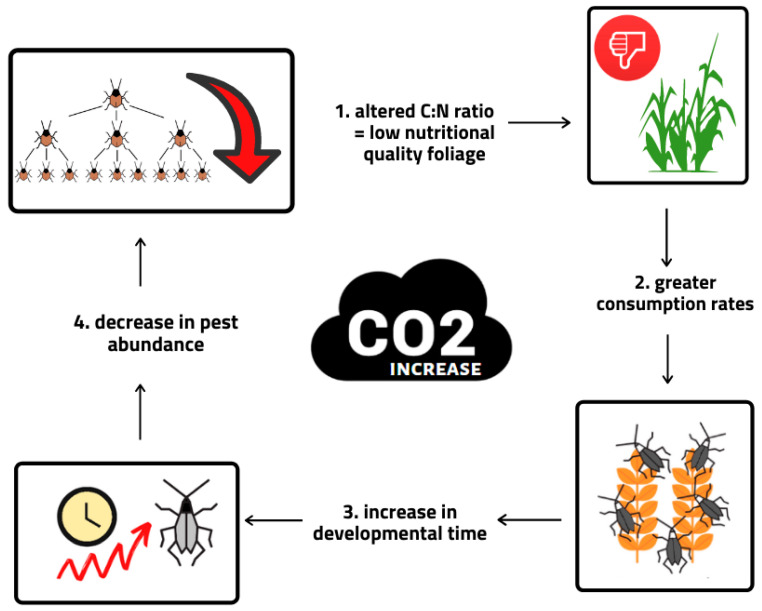
Impact of atmospheric CO_2_ increase on agricultural insect pests.

**Figure 3 insects-12-00440-f003:**
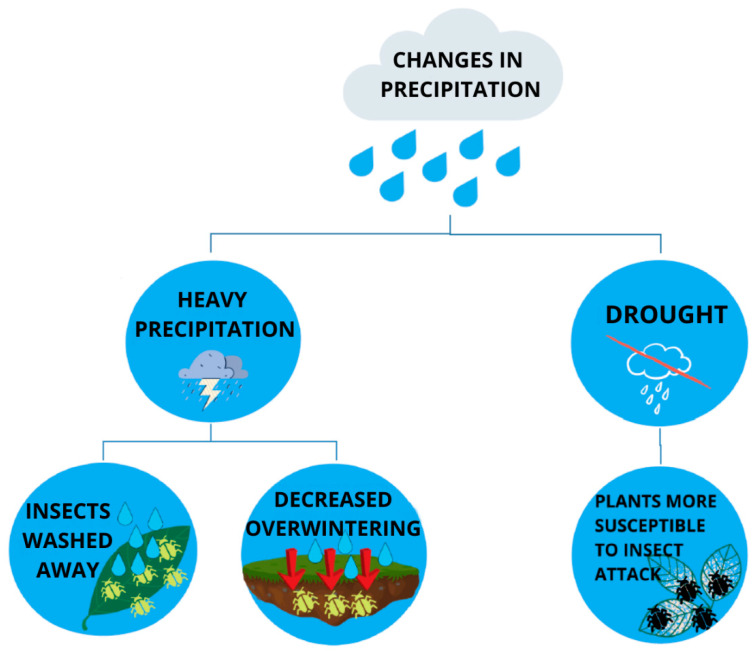
Impact of heavy precipitation and drought on agricultural insect pests.

**Figure 4 insects-12-00440-f004:**
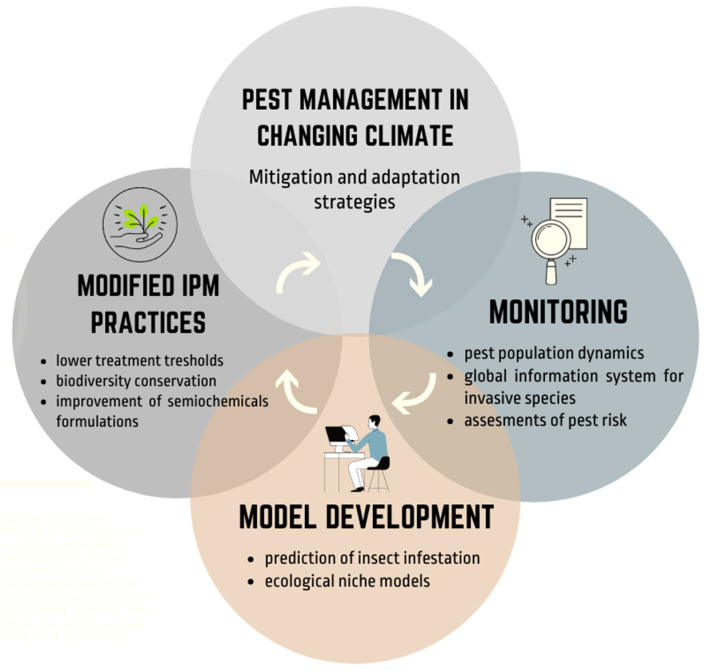
Potential pest management strategies for mitigation and adaption to new environmental conditions.

## Data Availability

No new data were created or analyzed in this study. Data sharing is not applicable to this article.

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
