# Peer review of "The Impact of Climate Change on Agricultural Insect Pests"

_insects, 2021, doi:10.3390/insects12050440_

Round 1

Reviewer 1 Report

This is a long comprehensive review of the effects of climate change on plants and insect pests. The authors do an excellent clearly explaining the impacts of climate change (warmer temperatures, higher CO2 levels, changes in precipitation, longer warmer seasons, etc.) and how this will impact plants and agricultural insects. The authors discuss how all these possible changes can be accounted for with modeling and how they must be considered in Integrated Pest Management plans.

This is a well written and easy to read and understand manuscript. The authors clearly lay out their argument and provide supporting evidence for it. This is a biased review, this is not meant to be a negative comment as this is not a systematic review and the format leads to bias in the papers that are selected. Therefore the evidence presented (papers cited) support the authors ideas, and papers that do not support them are not used. Although some sections do present unclear responses and cite papers showing both increases and decreases to pest populations.  Likely given the global diversity of insects, this uncertainty is species specific and the impact is unknown. There are trends and the authors do a good job explaining this, however in the conclusion a single line about how each pest may respond differently depending on the host plants, temperature, ect. may be needed to prevent people from drawing conclusions too quickly about specific insect species. Also the models and IPM parts seem significantly less comprehensive than the other sections.

The explanation about the impact of climate change on plants was unexpected in an insect journal. It was long and seemed out of place. However, it was clear, easy to follow, and I learned a lot. But was it needed to understand impact of insects in a changing climate? The authors used that background a few times such as talking about the lower nutritional value with increased CO2, and a few other times. Rather than asking for parts to be removed, can the authors refer back to the lessons taught in that section a bit more in the other sections specifically to the C3 and C4 aspects which are extensively explained but not really mentioned again in the manuscript.

The figures need to be improved. Although they graphically represent the texts, they are not informative and possibly not needed. Again, rather than remove them as they do serve a purpose, please make them more useful perhaps use a specific system to illustrate how it relates to the text. As they are shown, they do not build or add to the written text.

Minor comments, there is some repetition in the manuscript and that may be needed, but the authors often repeat the impacts of CO2 and temperature almost verbatim in each of the sections. I leave it to their discretion if this needs to be slightly reduced.

Author Response

Dear reviewer,

I would like to thank you for your time and effort in reading our article. Also, for your comments and suggestions.

RESPONSE TO REVIEWER’S COMMENTS:

Point 1. There are trends and the authors do a good job explaining this, however in the conclusion a single line about how each pest may respond differently depending on the host plants, temperature, ect. may be needed to prevent people from drawing conclusions too quickly about specific insect species. 

Response 1: Additional conclusion added in the text – now L1070 – L076

 Point 2. Also the models and IPM parts seem significantly less comprehensive than the other sections.

Response 2: Additional information and references added in the text – now L929 – L934, L949 – L957, L969 – L986

 Point 3. Rather than asking for parts to be removed, can the authors refer back to the lessons taught in that section a bit more in the other sections specifically to the C3 and C4 aspects which are extensively explained but not really mentioned again in the manuscript.

Response 3:  Additional information and references added in the text – now L312 – L315, L317 – L372

 Point 4. The figures need to be improved. Although they graphically represent the texts, they are not informative and possibly not needed. Again, rather than remove them as they do serve a purpose, please make them more useful perhaps use a specific system to illustrate how it relates to the text. As they are shown, they do not build or add to the written text.

Response 4: The figures have been made to illustrate the very complex sub-topic, and have been inserted in the manuscript exclusively for the purpose of clarification. If the editor suggests removing them, we will accept that decision.

 Point 5. Minor comments, there is some repetition in the manuscript and that may be needed, but the authors often repeat the impacts of CO2 and temperature almost verbatim in each of the sections. I leave it to their discretion if this needs to be slightly reduced.

Response 5:  We slightly reduced repeated parts in the text.

Reviewer 2 Report

This ms is a review on the effects of climate change on agricultural pests.

I think the authors have missed some very important papers on the topic, among them 

DeLucia EH, Casteel CL, Nabity PD, et al. 2008. Insects take a bigger
bite out of plants in a warmer, higher carbon dioxide world. P Natl
Acad Sci USA 105: 1781–82.

Deutsch CA, Tewksbury JJ, Huey RB, et al. 2008. Impacts of climate
warming on terrestrial ectotherms across latitude. P Natl Acad Sci
USA 105: 6668–72.

Deutsch CA, Tewksbury JJ, Tigchelaar M, et al. 2018. Increase in crop
losses to insect pests in a warming climate. Science 361: 916–19.

Lehmann P., Ammunét T., Barton M., Battisti A., Eigenbrode S.D., Jepsen J.U., Kalinkat G., Neuvonen S., Niemelä P., Terblanche J.S., Økland B., Björkman C. 2020. Complex responses of global insect pests to climate warming. Frontiers in Ecology and the Environment, 18, 141-150. doi: 10.1002/fee.2160

I invite the authors to read the papers above and come up with a critical analysis of the data available on the impacts of climate change on agricultural pests. Especially in Lehmann et al. they will find evidence of different responses to climate change in some main agricultural pests. The critical analysis should be presented in the conclusions, keeping in mind the high level of uncertainty associated with these predictions.

Author Response

Dear reviewer,

I would like to thank you for your time and effort in reading our article. Also, for your comments and suggestions.

 RESPONSE TO REVIEWER’S COMMENTS:

Point 1. I think the authors have missed some very important papers on the topic, among them 

DeLucia EH, Casteel CL, Nabity PD, et al. 2008. Insects take a bigger
bite out of plants in a warmer, higher carbon dioxide world. P Natl
Acad Sci USA 105: 1781–82.

Deutsch CA, Tewksbury JJ, Huey RB, et al. 2008. Impacts of climate
warming on terrestrial ectotherms across latitude. P Natl Acad Sci
USA 105: 6668–72.

Deutsch CA, Tewksbury JJ, Tigchelaar M, et al. 2018. Increase in crop
losses to insect pests in a warming climate. Science 361: 916–19.

Lehmann P., Ammunét T., Barton M., Battisti A., Eigenbrode S.D., Jepsen J.U., Kalinkat G., Neuvonen S., Niemelä P., Terblanche J.S., Økland B., Björkman C. 2020. Complex responses of global insect pests to climate warming. Frontiers in Ecology and the Environment, 18, 141-150. doi: 10.1002/fee.2160

I invite the authors to read the papers above and come up with a critical analysis of the data available on the impacts of climate change on agricultural pests. Especially in Lehmann et al. they will find evidence of different responses to climate change in some main agricultural pests. The critical analysis should be presented in the conclusions, keeping in mind the high level of uncertainty associated with these predictions.

 Response 1: References added in the text.

DeLucia EH, Casteel CL, Nabity PD, et al. 2008. Insects take a bigger
bite out of plants in a warmer, higher carbon dioxide world. P Natl
Acad Sci USA 105: 1781–82.

- reference added – now L261 – L264 and L331 – L336

Deutsch CA, Tewksbury JJ, Huey RB, et al. 2008. Impacts of climate
warming on terrestrial ectotherms across latitude. P Natl Acad Sci
USA 105: 6668–72.

- reference added – now L288 – L292

Deutsch CA, Tewksbury JJ, Tigchelaar M, et al. 2018. Increase in crop
losses to insect pests in a warming climate. Science 361: 916–19.

- reference added – now L264 – L276

Lehmann P., Ammunét T., Barton M., Battisti A., Eigenbrode S.D., Jepsen J.U., Kalinkat G., Neuvonen S., Niemelä P., Terblanche J.S., Økland B., Björkman C. 2020. Complex responses of global insect pests to climate warming. Frontiers in Ecology and the Environment, 18, 141-150. doi: 10.1002/fee.2160

- reference added – now L292 – L299

- conclusion added – now L070 – L076

Reviewer 3 Report

Manuscript ID Insects 1194818

The impact of climate change on agricultural insect pests

To Authors

A very nicely written review publication. It contains a wealth of literature and deals with climate change from many points of view. Reading is not interesting and does not bore the reader. The publication may be of assistance to scientists studying the aspects of climate change in the context of insect development. I accept it in present form.

Some other papers to add (corresponding to Integrated Pest Management):

  1. Tribolium confusum responses to blends of cereal kernels and plant volatiles. J. Appl. Entomol. 140: 558–563, (2016).

Author Response

Dear reviewer,

I would like to thank you for your time and effort in reading our article. Also, for your comments and suggestions.

RESPONSE TO REVIEWER’S COMMENTS:

Point 1. Some other papers to add (corresponding to Integrated Pest Management):

Tribolium confusum responses to blends of cereal kernels and plant volatiles. J. Appl. Entomol. 140: 558–563, (2016).

Response 1: Reference added in text – now L975 – L980.

Round 2

Reviewer 2 Report

The authors have thoroughly addressed my previous comments and the ms can now be accepted for publications.